# Effects of Beetroot Juice Supplementation on Cognitive Function, Aerobic and Anaerobic Performances of Trained Male Taekwondo Athletes: A Pilot Study

**DOI:** 10.3390/ijerph181910202

**Published:** 2021-09-28

**Authors:** Hossein Miraftabi, Zahra Avazpoor, Erfan Berjisian, Amir Sarshin, Sajjad Rezaei, Raúl Domínguez, Reid Reale, Emerson Franchini, Mohammad Hossein Samanipour, Majid S. Koozehchian, Mark E. T. Willems, Ramin Rafiei, Alireza Naderi

**Affiliations:** 1Department of Exercise Physiology, Faculty of Physical Education and Sport Sciences, Tehran University, Tehran 1417935840, Iran; miraftabi2018@ut.ac.ir (H.M.); avazpoor.z2015@ut.ac.ir (Z.A.); erfan.berjisian@ut.ac.ir (E.B.); ramin.rafiei73@gmail.com (R.R.); 2Clinical Care and Health Promotion Research Center, Karaj Branch, Islamic Azad University, Karaj 3149968111, Iran; amsarshin@gmail.com; 3Department of Physical Education & Sport Sciences, Faculty of Humanities, Tarbiat Modares University, Tehran 1411713116, Iran; rezaeisajjad@yahoo.com; 4Departamento de Motricidad Humana y Rendimiento Deportivo, Faculty of Education Sciences, Universidad de Sevilla, 41018 Sevilla, Spain; raul_dominguez_herrera@hotmail.com; 5Studies Research Group in Neuromuscular Responses (GEPREN), University of Lavras, Lavras 37200-000, Brazil; 6USA.UFC Performance Institute, Shanghai 200072, China; reid.reale@gmail.com; 7School of Physical Education and Sport, University of São Paulo, São Paulo 05508-030, Brazil; efranchini@usp.br; 8Department of Sport Science, Imam Khomeini International University, Qazvin 3414896818, Iran; Samani.mh@gmail.com; 9Department of Kinesiology, Jacksonville State University, Jacksonville, AL 36265, USA; mkoozehchian@jsu.edu; 10Institute of Sport, Nursing and Allied Health, College Lane, University of Chichester, Chichester PO19 6PE, UK; m.willems@chi.ac.uk; 11Department of Sport Physiology, Boroujerd Branch, Islamic Azad University, Boroujerd 6915136111, Iran

**Keywords:** combat sport, ergogenic aid, high-intensity intermittent performance, martial arts

## Abstract

Studies have shown that nitrate (NO_3_^−^)-rich beetroot juice (BJ) supplementation improves endurance and high-intensity intermittent exercise. The dose–response effects on taekwondo following BJ supplementation are yet to be determined. This study aimed to investigate two acute doses of 400 mg of NO_3_^−^ (BJ-400) and 800 mg of NO_3_^−^ (BJ-800) on taekwondo-specific performance and cognitive function tests compared with a placebo (PL) and control (CON) conditions. Eight trained male taekwondo athletes (age: 20 ± 4 years, height: 180 ± 2 cm, body mass: 64.8 ± 4.0 kg) completed four experimental trials using a randomized, double-blind placebo-controlled design: BJ-400, BJ-800, PL, and CON. Participants consumed two doses of BJ-400 and BJ-800 or nitrate-depleted PL at 2.5 h prior to performing the Multiple Frequency Speed of Kick Test (FSKT). Countermovement jump (CMJ) was performed before the (FSKT) and PSTT, whereas cognitive function was assessed (via the Stroop test) before and after supplementation and 10 min following PSTT. Blood lactate was collected before the CMJ tests immediately and 3 min after the FSKT and PSST; rating of perceived exertion (RPE) was recorded during and after both specific taekwondo tests. No significant differences (*p* > 0.05), with moderate and large effect sizes, between conditions were observed for PSTT and FSKT performances. In addition, blood lactate, RPE, heart rate, and CMJ height were not significantly different among conditions (*p* > 0.05). However, after the PSTT test, cognitive function was higher in BJ-400 compared to other treatments (*p* < 0.05). It was concluded that acute intake of 400 and 800 mg of NO_3_^−^ rich BJ reported a moderate to large effect size in anaerobic and aerobic; however, no statistical differences were found in taekwondo-specific performance.

## 1. Introduction

Taekwondo is an Olympic combat sport, with matches contested over three 2 min rounds separated by 1 min rest. It is a striking sport characterized by repeated high-intensity bursts of punching and predominantly kicking techniques lasting ~1–5 s, interspersed by low-intensity bouncing actions lasting ~10–35 s in a range from 1:2–1:7 work-to-rest ratios within each of the rounds [1]. Campos et al. [2] found that adenosine triphosphate (ATP) resynthesis during taekwondo competition is predominantly produced via the oxidative energy system (accounting for ~66 ± 6% of total energy cost). However, both anaerobic glycolytic and ATP-PCr systems contribute to energy cost for ~4 ± 2% and ~30 ± 6% of total energy expenditure, respectively. Oxidative metabolism is the primary metabolic pathway during taekwondo combats, but the contribution of non-oxidative metabolism is likely to be the main source supporting repeated high-intensity kicking actions [3], which are required to score points and win matches. Nutritional supplement strategies that may improve performance or recovery as well as aiding oxidative and/or non-oxidative metabolism relevant in combat sport are of interest to taekwondo athletes [4]. However, studies have investigated dietary supplements, such as vitamins [5,6,7], carbohydrate drinks [8], caffeine [9], creatine monohydrate [10] and sodium bicarbonate [3,10] in the context of taekwondo, but no studies have analyzed the possible ergogenic effect of nitrate (NO_3_^−^)-rich beetroot juice (BJ) supplementation. BJ is an ergogenic aid that has received increased attention in the last decade, garnering strong support for enhancing exercise performance [11]. NO_3_^−^ is a nitric oxide (NO) precursor and part of the NO_3_^−^—nitrite (NO_2_^−^)NO pathway [12], shown to modulate various aspects of physiological mechanisms, including muscle contraction and mechanical efficiency [13]. Improvements in exercise capacity are suggested to occur due to a lowering of the cost of ATP to exercising muscles [14,15], increased vasodilation [16], and mitochondrial respiration, as well as intracellular calcium handling [17]. In addition, BJ diminishes the reduction of phosphocreatine (PCr) for a determined work [18], a mechanism that may delay the depletion of PCr and favor the restoration of PCr during high-intensity intermittent efforts [19]. In addition, BJ supplementation may exert a central effect based on an increased level of tension and subjective vitality at the start of exercise [19,20] and improved physical performance with the same [21,22] or a lower [20,23] rating of perceived exertion (RPE). It is mediated by a possible increase in brain perfusion or a diminution of metabolite accumulation that attenuate activation of III/IV muscle afferent feedback to the central nervous system and the potential for central fatigue development [23]. A meta-analysis found that dietary NO_3_^−^ supplementation can improve endurance exercise performance [24] and improve cognitive function during exercises demanding quick and accurate decisions [25,26]. Additionally, increased power and force production in fast-twitch fibers [17,27] and improved reaction time [27,28] were reported with intake of BJ. As with all supplements, determining the ideal dose and duration of the intake protocol is necessary to optimize ergogenic benefits [29]. The International Olympic Committee (IOC) consensus statement suggests that an effective dosing protocol involves the acute ingestion of BJ in a range from 5 to 9 mmol (310–560 mg) dietary NO_3_^−^, 2–3 h pre-exercise [11], with lower dosages generally not showing beneficial effects on performance [30,31]. 

Taekwondo (and other combat sport) athletes may benefit from acute dietary NO_3_^−^ ingestion due to the intermittent nature of combat sports [19], and the physical fitness variables (such as strength, aerobic and anaerobic performance), and the cognitive functions required for successful performance in competition [1,32] Although positive effects of NO_3_^−^ supplementation on handgrip strength in jiu-jitsu athletes [33] have been reported, the ergogenic effects of dietary NO_3_^−^ ingestion on taekwondo or other striking combat sports performance are yet to be explored. In addition, the study that examined NO_3_^−^ supplementation on combat sports athletes utilized doses (12 mmol) well above that known to be generally efficacious in other contexts [33].

Accordingly, the present study aimed to examine the effect of two acute doses of NO_3_^−^ rich BJ (400 mg NO_3_^−^ vs. 800 mg NO_3_^−^) on aerobic and anaerobic taekwondo-specific performance, RPE, blood lactate, and cognitive function. The hypothesis was that the ingestion of a high dosage (800 mg NO_3_^−^) would increase taekwondo performance compared to a low dosage (400 mg NO_3_^−^) during specific taekwondo tests. However, high and low dosage of BJ supplementation would increase both cognitive function and exercise performance compared to placebo (PL) and control (CO) conditions.

## 2. Materials and Methods

### 2.1. Participants

The study started with 12 black belt male athletes from the Iranian National Taekwondo League, but four taekwondo athletes could not complete our study due to injury during the period of the investigation. Therefore, eight athletes (age: 20 ± 4 years; height: 180 ± 2 cm; body mass: 64.8 ± 4.0 kg) completed the study. All athletes were from the same club to prevent potential interference/variation caused by different training programs implemented among taekwondo clubs. The inclusion criteria included: (1) more than five years of experience in taekwondo; (2) not having consumed any ergogenic aids including: creatine, beta-alanine, caffeine, sodium bicarbonate, and caffeine content supplements three months before the study; (3) be training at least five times a week; (4) be older than 18 years old; (5) not currently having any musculoskeletal injuries. This study was conducted during the 4-week pre-competition preparation phase. During this time, participants trained six sessions per week, including three taekwondo-specific training and three conditioning sessions, including strength training and taekwondo-specific fitness. The Sport Science Research Institute of Iran approved of current study (IR.SSRC.REC.1399.062).

### 2.2. Experimental Design

This study had a randomized, double-blind placebo-controlled crossover design with five testing sessions at the laboratory. The first session was used to explain the experimental procedures, take anthropometric measurements, and familiarize with the Multiple Frequency Speed of Kick Test (FSKT), countermovement jump (CMJ), and the Progressive Specific Taekwondo Test (PSTT). All participants were familiar with the above specific tests, but under supervision of a qualified coach, the participants repeated the tests until the coach considered the technique was appropriate. Subsequently, participants were set to 4 conditions: (1) 60 mL BJ (400 mg NO_3_^−^) + 60 mL placebo (PL) (BJ-400), (2) 60 mL BJ + 60 mL BJ (BJ-800), (3) 60 mL PL + 60 mL PL (PL), and (4) control (CON). The washout between sessions was seven days, and all measurements were performed between 10:00 a.m. and 4:30 p.m. at the same day of each week to align with normal weekly routines [34]. Participants were instructed to prevent strenuous exercise 24 h before each session. To prevent overreaching, coaches were requested to control the training volume and intensity throughout the study. The Hooper Index questionnaire was used before each test to monitor and evaluate the recovery and accumulated fatigue [34,35]. The experimental design is summarized in Figure 1.

### 2.3. Supplementation Protocol and Standardization of Physical Activity and Diet

Participants arrived for the four trials (BJ-400, BJ-800, PL, CON) at the laboratory 3 h prior to the tests. Two hours and thirty minutes before starting the tests, each participant ingested either one bottle of 60 mL BJ + one bottle of 60 mL of PL or two bottles of BJ (120 mL) (Red Beet Vinitrox Shot; Sponsor Ltd.; Germany) containing 400 mg NO_3_^−^ per bottle or depleted dried powder NO_3_^−^ for PL [36]. Based on previous instructions, PL was prepared by dissolving 1 g of dried powder of BJ into one liter of water and adding lemon juice to mirror the taste of the commercial supplement [36]. Both BJ and PL were provided in identical bottles. Based on the pharmacodynamics of NO_2_^−^ after BJ ingestion [37] and the recommendation of ingesting BJ 2.5–3 h before exercise to synchronize with the peak of plasma NO_2_^−^ [11,29,38], supplementation was given 2.5 h previous to FSKT. In consideration of the anti-bacterial effect in the mouth by oral antiseptics which may prevent the rise in blood NO_2_^−^ after the intake of NO_3_^−^, participants were instructed to abstain from teeth brushing, using mouthwash, chewing gum, or sweets that could contain anti-bacterial substance during the 24 h prior to the test session [36]. Furthermore, a list of dietary sources rich in NO_3_^−^ was given to participants, with instructions to avoid eating those in the 24 h prior to arriving at the laboratory (i.e., beetroot, celery, arugula, lettuce, spinach, turnip, endives, leek, parsley, cabbage, etc.). Participants were also instructed to avoid drinks containing caffeine and nitrate-rich fruits and vegetables 72 h before the tests due to its potential ergogenic effect. Participants were asked to replicate their food and drink intake 24 h before each session. On the test day, water was provided ad libitum during the rest period between the tests, and a standardized snack was ingested four hours before starting the test consisted of 1.5 g/kg carbohydrate and 20 g protein [35].

### 2.4. Frequency Speed of Kick Test Multiple

The FSKT included 5 sets of FSKT, and each set’s period was 10 s interspersed with 10 s passive rest. Each athlete had to stay in front of the punching mitt to perform the test. After the sound signal, athletes performed the maximal number of kicks, alternating right and left legs. Performance was measured via the number of kicks in each set, the total number of kicks, and the kick decrement index (KDI) during the test. KDI indicates the degree that performance decreases during the test and was calculated based on the number of kicks performed during five sets of the FSKT [39] using the following equation:$$\mathrm{KDI}\;\left(\%\right)\;=\;1-\frac{\mathrm{FSKT}1+\mathrm{FSKT}2+\mathrm{FSKT}3+\mathrm{FSKT}4+\mathrm{FSKT}5}{\mathrm{Best}\;\mathrm{FSKT}\;\times \;\mathrm{Number}\;\mathrm{of}\;\mathrm{Sets}}\;\times \;100$$

### 2.5. Progressive Specific Taekwondo Test

The PSTT used a kick pad and was performed in a 2 m × 2 m area. Kicks had to be between the navel and nipples height. Participants started the PSTT from the first stage with six kicks, performed by the right leg alternating between right and left legs, and then progressively increasing by four kicks in each new stage. Athletes were standing the test in step (fighting stance hopping). Sound signals set the pace performance with the time distance interval between each kick fixed for each new stage, intervals decreased for every new stage, increasing the kicking frequency (more details are provided in the original reference) [40]. Each athlete continued to pace until exhaustion and received verbal encouragement such that participants performed the kicks with maximum power and maintained the technical quality of the kicks for the whole duration of the test. Heart rate (HR) was recorded (Polar H10, Kempele, Finland) every 5 s during the protocol. The HR at the end of the test was taken as HRpeak. During the PSTT, the highest frequency of kicks was defined as the peak frequency kick that athletes reached in the last stage. Athletes were instructed to perform the bandal-tchagui kick on the pad throughout the test. The test was stopped upon the athlete failing to maintain more than two standard technical kicks or when the athlete stopped the test [40].

### 2.6. Counter Movement Jump

Subjects had to perform a maximal CMJ via a validated cellphone app, the My jump 2 app [41]. Athletes performed the jump from a start position with extended knees, before squatting down to 90 knee flexion, with hands in mild pronation and without any subsequent arm swing, followed by an immediate jump for maximal height. Knees were extended during the flight stage, as arms moved down with the elbows extended, and contact with the ground was made with the toes. Athletes CMJ were recorded by the cellphone’s camera and My jump 2 app measured and calculate the CMJ components. This test was performed prior to the conduct of each specific taekwondo tests. Three CMJ were executed with a period of 30 s rest between jumps, and the highest jump was taken for analysis.

### 2.7. Lactate Measurement and Rating of Perceived Exertion Recording

Finger-prick blood samples for lactate were collected before the CMJ tests, immediately and 3 min after the PSST and FSKT (Lactate h/p/Cosmos, Germany). The rating of perceived exertion (RPE, 6-20 Borg scale) was recorded during and after the FSKT and PSST.

### 2.8. Cognitive Function Assessment

Cognitive function was assessed with the Stroop word–color test before and after supplementation and 10 min following the progressive specific taekwondo test [42]. The test has three pages including a word page printed with black words (W), a page containing “XXXX’’ printed in different colors (C), a final page combines “RED”, “GREEN” and “BLUE” words from the first page printed with colors from the second page, where the ink color did not match with text of the color word (CW). All subjects were instructed to respond as quickly as possible to each page for 45 s. The tests of the first two pages evaluated congruence, whereas the last test measures incongruence or interference to identify the true word without color effect on their choice to correct answer. The correct answers were scored for percentage accuracy, and reaction time was considered for assessing cognitive function.

### 2.9. Gastrointestinal Symptoms Assessment

Gastrointestinal (GI) symptoms after supplement ingestion were recorded with a gastrointestinal questionnaire [43]. Participants selected values ranging from 0 to 9, where 0 indicated “no problem at all” and 9 indicated “the worst it has ever been”. The symptoms were considered severe when the score was equal to or higher than 5.

### 2.10. Statistical Analysis

All results are reported as means and standard deviation. Normality was checked by using graphing (Boxplots, histograms, and scatterplots), interquartile range to create outlier fences, and hypothesis tests (Grubbs’ test) and all data were normally distributed (*p* > 0.05). Several one-way repeated measure ANOVA analyses were used to compare aerobic performance (TTE), anaerobic performance (total kicks and KDI), and height reached on CMJ between the four treatments. A two-way repeated measure (treatment-by-time) ANOVA analysis was used for each variable of anaerobic performance, Stroop test, lactate, and RPE. Bonferroni test was used as post hoc when a significant difference was found in the ANOVA. ANOVA-partial eta squared denoted as $\eta_{P}^{2}$ is the effect size (ES) for all types of ANOVA, which were classified as small (0.01), medium (0.06), and large effect (0.14 and higher). The statistical significance was set as *p* ≤ 0.05. Statistical analyses were performed using SPSS 22 (IBM Corp. Released 2013. IBM SPSS Statistics for Windows, Version 22.0. IBM Corp, Armonk, NY, USA) whereas figures were drawn using GraphPad Prism 8 statistical software (San Diego, CA, USA).

## 3. Results

### 3.1. Progressive Specific Taekwondo Test

For aerobic performance (TTE), there was a large ES with a trend of statistical differences between treatments (F_3,21_ = 2.79, *p* = 0.069, $\eta_{P}^{2}$ = 0.285) (Figure 2).

### 3.2. Kick Decrement Index and Frequency Speed of Kick Test Multiple

No differences were observed for anaerobic performance, but KDI had a large ES (F_3,21_ = 1.71, *p* = 0.195, $\eta_{P}^{2}$ = 0.196) (Figure 3), whereas total kicks in FSKT had a medium ES (F_3,21_ = 0.87, p = 0.471, $\eta_{P}^{2}$ = 0.111) (Figure 4). In addition, medium ES treatment-by-time interactions (F_12,84_ = 1.52, *p* = 0.134, $\eta_{P}^{2}$ = 0.178) were observed for the number of kicks in each set (F_3,21_ = 0.873, *p* = 0.471, $\eta_{P}^{2}$ = 0.111). The main effect of the set was significant (F_4,28_ = 35.04, *p* = 0.001, $\eta_{P}^{2}$ = 0.834), and the number of kicks decreased from set one to five in all treatments (*p* < 0.001), but there was no significant difference between sets four and five (*p* > 0.05) (Figure 5). 

### 3.3. Blood Lactate

There were no significant treatment-by-time interactions (F_15,105_ = 1.24, *p* = 0.252, $\eta_{P}^{2}$ = 0.151) and main effects of treatment for blood lactate (F_3,21_ = 1.05, *p* = 0.392, $\eta_{P}^{2}$ = 0.051). However, the main effect of time was significant (F_5,31_ = 31.01, *p* = 0.001, $\eta_{P}^{2}$ = 0.816), and it was lower in the pre-test of FSKT than the other five measurements (*p* < 0.001 for all comparisons). Blood lactate three minutes after FSKT was higher than before PSTT (*p* = 0.013) and three minutes after PSTT (*p* = 0.033) in all conditions, but there was no difference between conditions (Figure 6).

### 3.4. Counter Movement Jump

There were medium and large ES, but no significant differences between treatments before FSKT for average height (F_3,21_ = 1.073, *p* = 0.382, $\eta_{P}^{2}$ = 0.133), force output (F_3,21_ = 1.112, *p* = 0.366, $\eta_{P}^{2}$ = 0.137), highest jump height (F_3,21_ = 0.234, *p* = 0.872, $\eta_{P}^{2}$ = 0.032), power (F_3,21_ = 1.228, *p* = 0.324, $\eta_{P}^{2}$ = 0.149), velocity (F_3,21_ = 0.195, *p* = 0.898, $\eta_{P}^{2}$ = 0.027), and flight time (F_3,21_ = 0.253, *p* = 0.858, $\eta_{P}^{2}$ = 0.035). In addition, there were no significant differences between treatments before PSTT for average height (F_3,21_ = 2.004, *p* = 0.144, $\eta_{P}^{2}$ = 0.223), force output (F_3,21_ = 1.190, *p* = 0.338, $\eta_{P}^{2}$ = 0.145), highest jump height (F_3,21_ = 1.25, *p* = 0.317, $\eta_{P}^{2}$ = 0.151), power (F_3,21_ = 0.822, *p* = 0.497, $\eta_{P}^{2}$ = 0.105), velocity (F_3,21_ = 1.272, *p* = 0.310, $\eta_{P}^{2}$ = 0.154), and flight time (F_3,21_ = 1.833, *p* = 0.172, $\eta_{P}^{2}$ = 0.208) (see Table 1).

### 3.5. Cognitive Function

In the Stroop test, there were significant treatment-by-time interactions for word–color score (F_6,42_ = 4.530, *p* = 0.001, $\eta_{P}^{2}$ = 0.393), word score (F_6,42_ = 5.707, *p* = 0.001, $\eta_{P}^{2}$ = 0.449), and total score (F_6,42_ = 4.414, *p* = 0.002, $\eta_{P}^{2}$ = 0.387), but not for color score (F_6,42_ = 1.459, *p* = 0.216, $\eta_{P}^{2}$ = 0.172). Word–color score (*p* = 0.035), word score (*p* = 0.040), or total score (*p* = 0.020), with BJ-400 treatment being higher than control. Word–color score in post-ingestion of BJ-400 was lower than pre-ingestion (*p* = 0.018). Word–color score in post-test of control condition was higher than pre-ingestion (*p* = 0.046). The main effect of time was significant for color (F_6,42_ = 4.932, *p* = 0.024, $\eta_{P}^{2}$ = 0.413) and increased from pre-ingestion to post-test in all experimental conditions (see Table 2).

### 3.6. Rating of Perceived Exertion

There were moderate and large ES, but no significant treatment-by-time interactions for RPE in five sets of FSKT (F_12,84_ = 1.19, *p* = 0.289, $\eta_{P}^{2}$ = 0.146) (Table 3) and levels of PSTT (F_36,36_ = 0.82, *p* = 0.724, $\eta_{P}^{2}$ = 0.450) (Table 4). The main effects of treatment for PSTT (F_3,3_ = 1.663, *p* = 0.343, $\eta_{P}^{2}$ = 0.624), and FSKT (F_3,21_ = 2.508, *p* = 0.087, $\eta_{P}^{2}$ = 0.264) were not significant different. However, the main effect of time was significant in FSKT (F_4,28_ = 105.443, *p* = 0.001, $\eta_{P}^{2}$ = 0.938), and PSTT (F_12,12_ = 542.888, *p* = 0.001, $\eta_{P}^{2}$ = 0.998).

### 3.7. Heart Rate

No differences were detected between treatments for HRpeak (F_3,21_ = 2.76, *p* = 0.068, $\eta_{P}^{2}$ = 0.283) and average HR (F_3,21_ = 1.10, *p* = 0.372, $\eta_{P}^{2}$ = 0.135) for FSKT, but with a medium or large ES (see Table 5). In addition, HRpeak (F_3,21_ = 0.76, *p* = 0.531, $\eta_{P}^{2}$ = 0.098) and average HR (F_3,21_ = 0.67, *p* = 0.579, $\eta_{P}^{2}$ = 0.087) for PSTT did not differ between conditions (Table 5).

### 3.8. Gastrointestinal Symptoms

Three (37.5%) participants in BJ-800 and two (25%) in BJ-400 reported minor belching problems. One (12.5%) participant in BJ-800 and one (12.5%) in BJ-40 reported very, very minor bloating problems. One (12.5%) participant in BJ-800 reported a moderate bloating problem. In BJ-800, one (12.5%) participant reported very, very minor stomach pain and cramps problem, one (12.5%) very minor stomach pain and cramps problem, two (25%) moderate stomach pain and cramps problem, and one (12.5%) very severe stomach pain and cramps problem. In BJ-400, one (12.5%) participant reported very minor stomach pain and cramps problems, and one (12.5%) severe problem. In BJ-800, one (12.5%) participant reported a very minor nausea problem and two (25%) very minor nausea problems. In BJ-400, one (12.5%) participant reported very, very minor nausea problems, two (25%) very minor nausea problems, and one (12.5%) moderate nausea problems. In BJ-800, one (12.5%) participant, and in the BJ-400 one (12.5%) reported very minor diarrhea problems. In BJ-800, one (12.5%) participant reported minor heartburn problems, one (12.5%) moderate heartburn problem, and one (12.5%) very, very severe heartburn problem. In PLA and CON, none of the participants reported GI discomforts.

## 4. Discussion

The primary aim of the current study was to assess the acute effects of two different dosing strategies of BJ containing 400 mg (BJ-400) and 800 mg of NO_3_^−^ (BJ-800) on taekwondo-specific performance. The results failed to support the original hypothesis and showed no differences in taekwondo-specific performance when acute doses of NO_3_^−^ rich in BJ were compared to PLA and CON conditions. However, BJ400 improved cognitive function compared to other treatments. 

Theoretically, BJ supplementation can improve the performance of high-intensity exercise modalities relying on fast-twitch muscle fiber recruitment [19]. BJ supplementation increases blood flow [44], vascular conductance [45], intracellular calcium handling [17], and may delay progressive depletion of PCr [19]. To determine if dietary NO_3_^−^ would improve specific anaerobic performance of taekwondo athletes, we analyzed the effect of 400 mg NO_3_^−^ (BJ-400) and 800 mg NO_3_^−^ (BJ-800) on FSKTmult (which consists of 5 sets of 10 s of maximum effort interspersed with 10 s passive rest). Contrary to the original hypothesis, the present study observed that neither BJ-400 nor BJ-800 increased the number of kicks in each set, the total number of kicks, or KDI compared with PLA and CON. Despite these consistent results, the previous studies examining the effects of NO_3_^−^ supplementation on high-intensity intermittent exercise performance have been equivocal [19,30,36,44,46,47,48,49,50,51]. A variety of dosages and intake durations, test protocols, exercise modalities, and work-to-rest ratios have been implemented, making difficult a direct comparison to the current study. Nevertheless, all of the studies showing an ergogenic effect of BJ supplementation on high-intensity intermittent exercise performance following chronic strategy (>2 days) [26,44,49,50]. In addition to our study, Muggeridge et al. [46] and Martin et al. [47] detected no ergogenic effect of acute BJ supplementation on high-intensity exercise performance. Muggeridge et al. [46], specifically, analyzed the effect of a single nitrate-rich concentrated BJ (5 mmol NO_3_^−^) taken 180 min before on an interval kayaking sprint test (five 10 s all-out sprints interspersed with 50 s of active recovery on kayak ergometer) [46]. The authors reported no statistical differences in either power output, total work, or fatigue index [46].

Therefore, we speculate that the acute supplementation protocol may not be sufficient to take advantage of BJ on high-intensity intermittent exercise performance. Hernandez et al. [17] reported an increase in force development and the expression of the sarcoplasmic reticulum (Ca^2+^) handling proteins only in type II muscle fibers of mice after seven days of NO_3_^−^ supplementation. An improvement in rats’ oxygenation of type II muscle fibers after five days of BJ consumption has also been reported [45]. Although these potentially ergogenic mechanisms following chronic BJ supplementation remained to be confirmed in humans, the data to date suggest a chronic loading of NO_3_^−^ is required to elicit skeletal muscle adaptations mainly responsible for high-intensity intermittent exercise performance [52]. Beyond the duration of supplementation, it is evident that there are large variations in NO_3_^−^ levels of commercial products [53,54] and between different samples of the same product [53]. Wruss et al. reported that 60 mL of the Red Beet Vinitrox Shot corresponds to 240 mg nitrate per serving while the company claims 400 mg nitrate per serving [54]. Our participants consumed one or two shots of 60 mL Red Beet (120 mL). Nevertheless, several subjects in the 120 mL BJ (BJ-800) condition had GI problems and did not experience ergogenic effects. In addition, a correlation was reported between the increase in plasma NO_2_^−^ following NO_3_^−^ supplementation and improved performance [37]. Therefore, the absence of assessing NO_2_^−^ or NO levels in the present study is a limitation because it was not possible to determine whether NO_2_^−^ levels were increased sufficiently for obtaining ergogenic effects. 

In the past years, nitrate-related research has shown the ergogenic effects of BJ supplementation on exercise performance and aerobic capacity in endurance athletes [55,56]. Thus, it was hypothesized that BJ supplementation could have beneficial effects on taekwondo athletes during the PSTT test. However, our results indicated that BJ-400 and BJ-800 supplementation could not improve PSTT compared with PLA and CON. To the best of our knowledge, the current findings are the first evidence to assess specific aerobic performance in taekwondo athletes following BJ supplementation. Therefore, a direct comparison between these findings and other taekwondo-specific studies is not possible. In contrast to our results, previous studies with other sports and exercise modes reported improvements for the incremental cycle test [57] and the 4 km time trial by NO_3_^−^ supplementation [58]. Kelly et al. [55] reported improvement with BJ (8.2 mmol NO_3_^−^) at intensities of 60%, 70%, and 80% peak power during incremental tests. Conversely, several studies reported no significant improvement in trained runners during the incremental running time to exhaustion (TTE) test [59], cycling time-trial performance [60], 40 min cycle ergometer distance-trial test [61], and 5 km running time-trial performance [62] following BJ supplementation. Inconsistent results related to the ergogenic effects of BJ supplementation on exercise performance may be due to the athletes’ fitness levels. As such, a study showed that ingestion of BJ did not reduce running VO2 or improve 1500 m time-trial performance in a group of elite distance runners following acute and chronic dietary NO_3_^−^ supplementation [63], whereas BJ reduced 2000 m time in recreational male and female runners [23]. A higher energetic demand and higher intake of dietary NO_3_^−^ in trained athletes compared to non-athlete individuals are the main factors that may limit the ergogenic effect of dietary NO_3_^−^ supplementation in trained athletes [64]. This may explain the lack of ergogenic response in well-trained athletes. Furthermore, it has been demonstrated that a high-intensity training program, which is a training strategy used by taekwondo athletes, may result in physiological adaptations (e.g., induced vasodilation) in these athletes, which may reduce NO action during exercise [65]. Therefore, BJ supplementation’s lack of positive effect on exercise performance observed in the current study may be attributed to the physiological adaptations occurring due to the high-intensity intermittent training by taekwondo athletes [33].

We observed no effect of supplementation on CMJ variables. Our results were consistent with studies showing no improvement in CMJ performance following BJ consumption [19,66,67,68]. NO_3_^−^ supplementation appears to be less effective for explosive actions such as CMJ although it seems that factors, including habitual consumption (i.e., daily NO_3_^−^ intake), and exercise type, may affect CMJ variables [66]. For instance, in terms of exercise type, it was reported that NO_3_^−^ supplementation has larger effects in exercise with involvement of small muscle, compared with whole-body or large muscle exercise [66]. There were significant increases in certain cognitive measures (color and word–color scores) in BJ-400 compared to other treatments. These findings agree with previous studies investigating the benefits of NO_3_^−^ supplementation on cognitive function [26]. However, it is unclear why BJ-800 had no effects on cognitive function in this study. This improvement in BJ-400 condition might be a consequence of enhanced cerebral oxygenation due to NO_3_^−^ supplementation [25]. Besides, previous findings have shown that high-intensity intermittent exercise may improve cognitive function [69,70]. Thus, more investigation is needed to determine the possible impact of BJ supplementation on cognitive function. 

There was no difference in RPE between conditions during sets of FSKT and PSTT. Our results agree with observations reported in the literature, which shows that BJ supplementation does not have a positive effect on RPE [67,68]. Our findings demonstrated no statistical improvement in peak and average HR during the FSKT and PSTT tests following BJ supplementation. Although most available literature describing the advantages of NO_3_^−^ on HR in heart failure patients [71], our results are consistent with previous studies that reported no effect of BJ on HR in healthy athletes [72,73]. As expected, during high-intensity exercise, blood lactate was increased three minutes after FSKT and PSTT in all conditions. However, the lactate concentrations were not different between conditions. These findings corroborate previous studies that described a lack of positive effects BJ supplementation on lactate concentration after high-intensity intermittent exercise [50]. A study reported chronic NO_3_^−^ supplementation during a four-week training program caused a lowering in blood lactate during exercise in recreational runners [74]. This might have been due to a change in the energy supply from an anaerobic source to an oxidative supply, as suggested by Wylie et al. [75]. These data show that the benefits of NO_3_^−^ supplementation on blood lactate may depend on the supplementation period/protocol, and athletes’ physical fitness level [36,74]. GI discomforts sometimes attribute to intake of nutritional supplements, negatively affecting the quality of life and athletic performance. Nevertheless, in our study, we reported a moderate to severe side effect in five participants including: bloating stomach pain and cramps, heartburn, and heartburn following ingestion of BJ-800 supplement. Two participants also reported severe GI discomfort and moderate nausea problem separately in BJ-400 condition. These events are possibly related to the content of commercial beetroot products. Commercial beetroot products contain various compounds (i.e., minerals, betalains, oxalic acid, phenolic acids, and sugars), of which ingestion of large volumes of these compounds might provoke gastric discomfort [54,76].

The high level of performance of the participants (member of the Iranian National Taekwondo League) gave rise to a small sample size that may mask ergogenic effects of BJ. In this sense, however, statistical differences in the physical performance variables were not reported, but we did report a trend for statistical differences in aerobic TTE performance and a moderate or large ES in aerobic and anaerobic performances. Considering that it is difficult to detect statistical differences in small sample sizes [77] and that minimal differences in performance (<0.6%) is important for competition [78], the effects of BJ could provide important practical implications for competitive athletes as participants in this study. In addition to the sample size, in this study, the NO_2_^−^ and NO concentrations were not measured. Therefore, based on the relationship between the magnitude of the increased plasma levels and increased performance, the use in this study of a supplement that could have lower levels of NO_3_^−^ than declared by the manufacturer [54] cannot ensure that NO_2_^−^ and NO levels increased more than the amount necessary for reaching ergogenic effects. All these limitations suggest that these results must be considered as caution. Thus, future studies must analyze the effect of BJ on taekwondo athletes, including a larger sample size, analyzing the NO_3_^−^ amount ingested by athletes and the increased plasma NO_2_^−^ and NO level post-supplementation for correlating differences in plasma value with differences in sport performance. In addition, future studies may also want to address the effects of acute and chronic supplementation in taekwondo athletes.

## 5. Conclusions

In summary, the results of this pilot study indicated that an acute BJ ingestion containing 400 mg of NO_3_^−^ improved cognitive function. However, the ingestion of acute two doses of BJ supplementation containing 400 and 800 mg of NO_3_^−^ in trained taekwondo athletes provided moderate and large effect sizes on anaerobic and large effect size on aerobic, but not statistical differences in taekwondo-specific performance.

## Figures and Tables

**Figure 1 ijerph-18-10202-f001:**
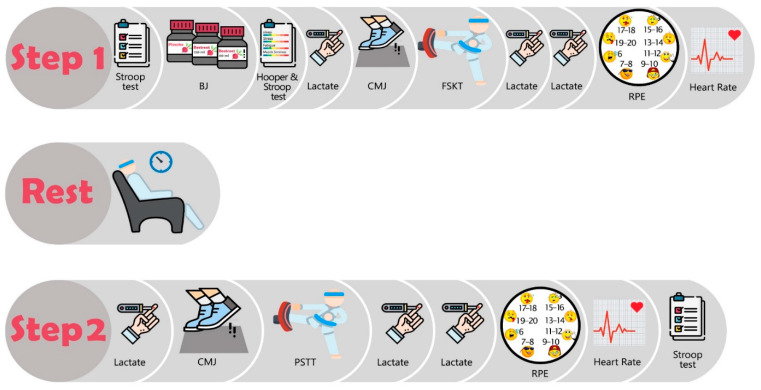
Schematic representation of the experimental design. BJ, beetroot juice; CMJ, counter movement jump; FSKT, Frequency Speed of Kick Test; PSTT, Progressive Specific Taekwondo Test; RPE, rating of perceived exertion.

**Figure 2 ijerph-18-10202-f002:**
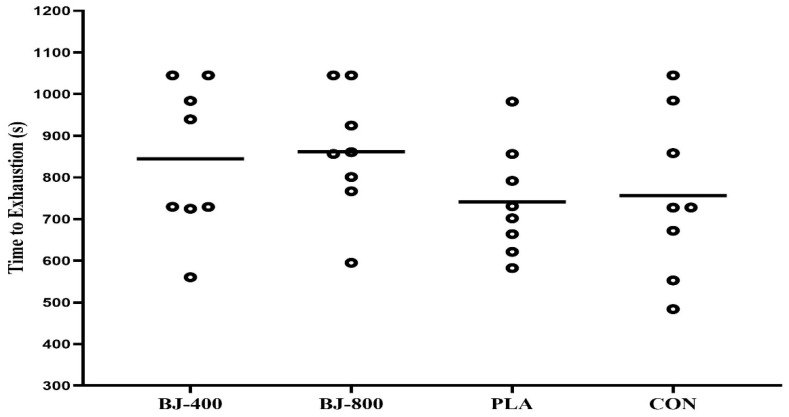
Time to exhaustion during the Progressive Specific Taekwondo Test in four treatments. BJ, beetroot juice; PLA, placebo; CON, control.

**Figure 3 ijerph-18-10202-f003:**
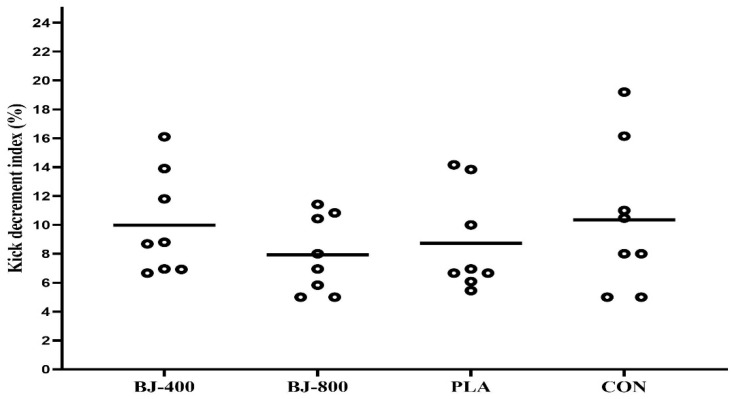
The kick decrement index in five sets of Frequency Speed of Kick Test in four treatments. BJ, beetroot juice; PLA, placebo; CON, control.

**Figure 4 ijerph-18-10202-f004:**
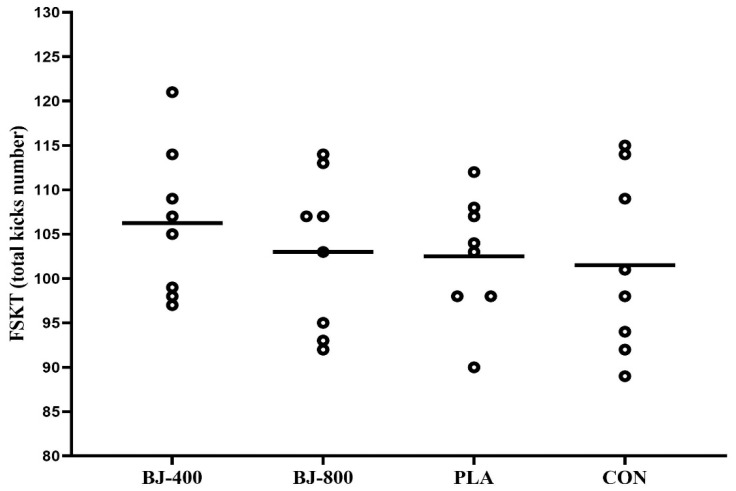
The number of total kicks in five sets of Frequency Speed of Kick Test in four treatments. BJ, beetroot juice; PLA, placebo; CON, control.

**Figure 5 ijerph-18-10202-f005:**
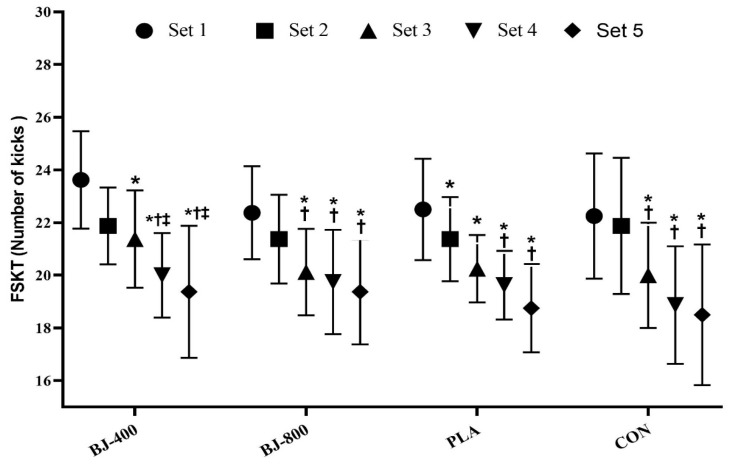
The number of kicks in five sets of Frequency Speed of Kick Test. Values are presented as mean and standard deviation. BJ, beetroot juice; PLA, placebo; CON, control. * Significant difference with Set 1 in same treatment (*p* < 0.05). † Significant difference with Set 2 in same treatment (*p* < 0.05). ‡ Significant difference with Set 3 in same treatment (*p* < 0.05).

**Figure 6 ijerph-18-10202-f006:**
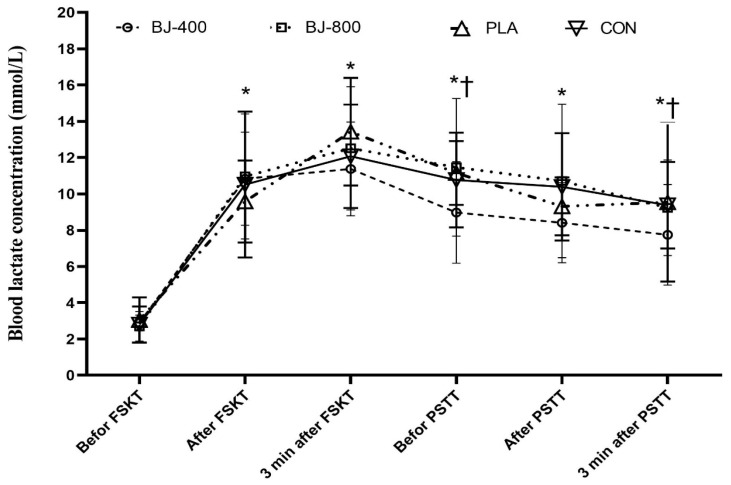
Blood lactate before and after Frequency Speed of Kick Test and Progressive Specific Taekwondo Test. Values are presented as mean and standard deviation. BJ, beetroot juice; PLA, placebo; CON, control. * Significant difference before FSKT in all treatments (*p* < 0.05). † Significant difference 3 min after FSKT in all treatments (*p* < 0.05).

**Table 1 ijerph-18-10202-t001:** Countermovement jump performance before the different experimental conditions.

	Before FSKT BJ-400	Before PSTT BJ-400	Before FSKT BJ-800	Before PSTT BJ-800	Before FSKT PLA	Before PSTT PLA	Before FSKT CON	Before PSTT CON
Highest Jump (cm)	42.9 ± 4.9	41.7 ± 4.9	42.3 ± 5.1	41.2 ± 4.7	42.3 ± 4.4	40.4 ± 4.3 *	42.7 ± 3.1	39.8 ± 2.8 **
Average Height (cm)	42.0 ± 4.7	40.6 ± 4.9	40.6 ± 4.6	38.9 ± 4.9	40.9 ± 3.4	39.3 ± 4.2 *	41.5 ± 3.1	38.5 ± 3.1 **
Flight Time (ms)	590.7 ± 30.9	583.5 ± 34.5	586.4 ± 34.6	579.0 ± 32.9	586.6 ± 29.7	570.9 ± 33.4 *	590.0 ± 21.6	564.4 ± 30.5 **
Velocity (m·s^−1^)	1.441 ± 0.078	1.431 ± 0.086	1.439 ± 0.086	1.410 ± 0.085	1.438 ± 0.071	1.405 ± 0.072	1.450 ± 0.058	1.395 ± 0.049 **
Force (*n*·kg^−1^)	1.435 ± 0.134	1.414 ± 0.110	1.426 ± 0.158	1.360 ± 0.127	1.415 ± 0.097	1.377 ± 0.062 *	1.499 ± 0.175	1.445 ± 0.154 **
Power (W·kg^−1^)	2.056 ± 0.287	2.010 ± 0.197	2.059 ± 0.320	1.996 ± 0.245	2.011 ± 0.209	1.909 ± 0.143 *	2.168 ± 0.295	2.001 ± 0.223

* Significant difference with before FSKT in PLA (*p* < 0.05). ** Significant difference with before FSKT in CON (*p* < 0.05); BJ, beetroot juice; PLA, placebo; CON, control. Values are presented as means and standard deviation.

**Table 2 ijerph-18-10202-t002:** Stroop test performance in the different experimental conditions.

		Word–Color (*n*)	Word (*n*)	Color (*n*)	Total (*n*)
BJ-400	Pre-ingestion	135 ± 11	149 ± 10	88 ± 2	373 ± 21
	Post-ingestion	119 ± 16	134 ± 20	96 ± 15	349 ± 48
	Post-exercise	130 ± 19	135 ± 20	97 ± 17	362 ± 55
BJ-800	Pre-ingestion	119 ± 16	131 ± 12	91 ± 11	342 ± 38
	Post-ingestion	117 ± 18	128 ± 17	93 ± 14	339 ± 46
	Post-exercise	125 ± 18	136 ± 20	99 ± 16	361 ± 52
PLA	Pre-ingestion	117 ± 14	127 ± 16	93 ± 7	338 ± 32
	Post-ingestion	116 ± 13	127 ± 10	92 ± 8	335 ± 29
	Post-exercise	124 ± 9	133 ± 6	102 ± 13	360 ± 25
CON	Pre-ingestion	114 ± 11	122 ± 15	87 ± 6	324 ± 32
	Post-ingestion	121 ± 15	128 ± 15	93 ± 14	343 ± 43
	Post-exercise	124 ± 17	136 ± 20	101 ± 20	362 ± 56

Values are presented as mean and standard deviation. BJ, beetroot juice; PLA, placebo; CON, control.

**Table 3 ijerph-18-10202-t003:** Rating of perceived exertion after each set of the Frequency of Speed Kick Test.

	Set 1 (a.u.)	Set 2 (a.u.)	Set 3 (a.u.)	Set 4 (a.u.)	Set 5 (a.u.)
BJ-400	6 ± 1	8 ± 2	11 ± 2	12 ± 2	14 ± 2
BJ-800	7 ± 1	9 ± 1	11 ± 1	12 ± 2	13 ± 1
PLA	7 ± 1	10 ± 1	12 ± 2	14 ± 2	15 ± 2
CON	6 ± 1	8 ± 1	10 ± 2	12 ± 2	13 ± 2

Values are presented as mean and standard deviation. a.u., arbitrary unit; BJ, beetroot juice; PLA, placebo; CON, control.

**Table 4 ijerph-18-10202-t004:** Rating of perceived exertion (RPE) in each level of the Progressive Specific Taekwondo Test and the number of participants reaching the level.

Treatment	Level 1	Level 2	Level 3	Level 4	Level 5	Level 6	Level 7	Level 8	Level 9	Level 10	Level 11	Level 12	Level 13	Level 14	Level 15
BJ-400 RPE *n*	6 ± 1 8	7 ± 12 8	8 ± 1 8	9 ± 2 8	9 ± 3 8	10 ± 3 8	11 ± 3 8	12 ± 3 8	13 ± 3 7	15 ± 3 7	15 ± 2 4	15 ± 1 4	17 ± 1 4	18 ± 1 3	18 ± 1 2
BJ-800 RPE *n*	6 ± 1 8	7 ± 1 8	9 ± 1 8	10 ± 2 8	11 ± 2 8	12 ± 2 8	14 ± 2 8	14 ± 1 7	16 ± 1 7	16 ± 1 6	17 ± 1 6	17 ± 1 6	18 ± 1 3	19 ± 0 2	19 ± 1 2
PLA RPE *n*	6 ± 1 8	8 ± 1 8	9 ± 2 8	10 ± 2 8	11 ± 2 8	13 ± 1 8	14 ± 2 8	15 ± 1 8	16 ± 1 6	17 ± 1 5	18 ± 1 4	19 ± 1 2	19 ± 1 2	20.0 1	-
CON RPE *n*	6 ± 1 8	8 ± 1 8	9 ± 1 8	10 ± 2 8	12 ± 2 8	13 ± 3 8	14 ± 3 8	15 ± 3 7	16 ± 3 6	16 ± 2 5	18 ± 2 4	18 ± 2 3	17 ± 2 2	18 ± 1 2	19 1

**Table 5 ijerph-18-10202-t005:** Heart rate responses to Frequency of Speed Test and Progressive Specific Taekwondo Test.

	FSKT		PSTT	
	HR_peak_ (bpm)	HR_average_ (bpm)	HR_max_ (bpm)	HR_average_ (bpm)
BJ-400	185 ± 7	174 ± 11	194 ± 7	174 ± 11
BJ-800	188 ± 5	169 ± 9	195 ± 7	174 ± 5
PLA	190 ± 5	175 ± 7	191 ± 12	178 ± 10
CON	190 ± 8	175 ± 7	194 ± 6	175 ± 8

## Data Availability

The current data in this study are available on request from the corresponding author.

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
