# Peer review of "Effects of Beetroot Juice Supplementation on Cognitive Function, Aerobic and Anaerobic Performances of Trained Male Taekwondo Athletes: A Pilot Study"

_ijerph, 2021, doi:10.3390/ijerph181910202_

Round 1

Reviewer 1 Report

I commend the authors for conducting this study in an elite group of athletes. 

Ln 128 – explain this familiarization session with more detail. How many attempts did the athletes do of each test? Were the athletes familiar with these tests prior to this familiarization day?  Please provide references to validate the use of the FSKT and PSTT for testing purposes. What is the typical CV between tests for each athlete?

Ln 132 – please make it clear whether athletes completed their individual trials at the same tiem each week.

Figure 1 – although this schematic is optically appealing, it is confusing as the pictures don’t completely align with the writing.  I believe this needs to be changed to be more clear and possibly consider making it more scientific in nature.

Ln 134 – how did the coaches control for the training load the day prior to test days and the day of test days?  Did athlete restrain from training for 12-24 hrs prior to testing? Please state this.

Ln 175 – did athletes replicate their diet before and on test day?

Ln 175 – how did you control for hydration status?

Ln 200-213 – did you calculate the athlete’s CV for each test and factor this into the results? 

I believe your results would be better displayed if you assessed based on magnitude-based inferences and individual responsiveness to the BRJ.  Are the results significant from an athlete performance standpoint and not from traditional statistics and p-values?  This is where assessing the CV for each athlete and looking at individual responsiveness to the BRJ is most important to determine practical significance.

Do you believe the athletes needed longer time (i.e. 180 minutes not 150 minutes) to see the peak in NO3- in the blood? Why did you ingest only 150 minutes prior to the tests and not wait the entire 180 minutes?

Author Response

I commend the authors for conducting this study in an elite group of athletes. 

Author Responds:

Many thanks for your attention to our study. We appreciate all your comments that we have answered point by point. We believe that your suggestions have improved the quality of the current version of the manuscript.

Ln 128 – explain this familiarization session with more detail. How many attempts did the athletes do of each test? Were the athletes familiar with these tests prior to this familiarization day?  Please provide references to validate the use of the FSKT and PSTT for testing purposes. What is the typical CV between tests for each athlete?

Author Responds: We have revised the familiarization and explained it with more details.

Line 155-160: The first session was used to explain the experimental procedures, taking anthropometric measurements, familiarization with the Multiple Frequency Speed of Kick Test (FSKT), countermovement jump (CMJ), and the Progressive Specific Taekwondo Test (PSTT). All participants were familiar with the above specific tests under supervise of a qualified coach and repeated the tests until the coach accept the correct technique. 

Ln 132 – please make it clear whether athletes completed their individual trials at the same tiem each week.

Author Responds: Line 165-169 -The washout between sessions was seven days, and all evaluations were done between 10:00 a.m. and 4:30 p.m. at the same time of the day for each participant, and at at the same day of each week to align with normal weekly routines [28]

Figure 1 – although this schematic is optically appealing, it is confusing as the pictures don’t completely align with the writing.  I believe this needs to be changed to be more clear and possibly consider making it more scientific in nature.

Author Responds: we have improved it based on your suggestion

Ln 134 – how did the coaches control for the training load the day prior to test days and the day of test days?  Did athlete restrain from training for 12-24 hrs prior to testing? Please state this.

Author Responds: we have revised it.

line169-172: To prevent overreaching, coaches were requested to control training volume and intensity throughout the study. The Hooper Index questionnaire was used before each test to monitor and evaluate the recovery and accumulated fatigue [29].

Ln 175 – did athletes replicate their diet before and on test day?

Author Responds: Yes, athletes replicate their diet before and on test day.

line219-225: Participants were also instructed to avoid drinks containing caffeine and nitrate-rich fruits and vegetables 72 h before the tests due to its potential ergogenic effect. Participants were asked to re-plicate their food and drink intake 24 h before each session. On the test day, water was provided ad libitum during the rest period between the tests and a standardized snack was ingested four hours before starting the test consisted of 1.5 g/kg carbohydrate and 20-gram protein.

Ln 175 – how did you control for hydration status?

Author Responds:  it was added. please, check lines 223.

Ln 200-213 – did you calculate the athlete’s CV for each test and factor this into the results? 

Author Responds: Many thanks, but we don’t think this approach would be an appropriate method to analyze as we measured performance by two specific taekwondo tests but not a simulated effort.

I believe your results would be better displayed if you assessed based on magnitude-based inferences and individual responsiveness to the BRJ.  Are the results significant from an athlete performance standpoint and not from traditional statistics and p-values?  This is where assessing the CV for each athlete and looking at individual responsiveness to the BRJ is most important to determine practical significance.

Author Responds:

Thank you very much with this suggestion. We have reflected reflected about this so much. We appreciate your suggestion and suggestion of reviewer 3# who proposed an alternative interpretation of the results. Considering that MBI is a method that currently have been under heavy criticism (i.e., https://doi.org/10.1111/sms.13491), finally we decided to do the suggestions of reviewer 3. We hope you consider this new version of the manuscript more appropriate.

Do you believe the athletes needed longer time (i.e. 180 minutes not 150 minutes) to see the peak in NO3- in the blood? Why did you ingest only 150 minutes prior to the tests and not wait the entire 180 minutes?

Author Responds: we have chosen this time because the duration of the experimental protocol was ~30 minutes. Adjusting the intake 150 minutes previous to the start of the experimental session, the effort was adjusted to a range from 150 to 180 minutes post-supplementation. In this sense and considering a previous study (Wylie, 2013) when a dosage of 8.4 mmol NO3- (~512 mg), an amount understand between the 2 dosages used on this study) was obtained 2 hours post-ingestion for reflexing a slight reduction at 4 hours post-ingestion. Considering the duration of the study and the results of Wylie et al. (2013), we thought that participants were on an optimal framework during all the experimental session. In addition, we ensured to be in the optimal range located on the range recommended by two of the most important organizations on sport nutrition, IOC (Maughan, 2018) and ISSN (Stecker, 2019) that recommend the timing of beetroot ingestion between 150 -180 min prior to exercise.

Wylie LJ, Kelly J, Bailey SJ, Blackwell JR, Skiba PF, Winyard PG, Jeukendrup AE, Vanhatalo A, Jones AM. Beetroot juice and exercise: pharmacodynamic and dose-response relationships. J Appl Physiol (1985). 2013 Aug 1;115(3):325-36. doi: 10.1152/japplphysiol.00372.2013.

Maughan RJ, Burke LM, Dvorak J, Larson-Meyer DE, Peeling P, Phillips SM, Rawson ES, Walsh NP, Garthe I, Geyer H, Meeusen R, van Loon LJC, Shirreffs SM, Spriet LL, Stuart M, Vernec A, Currell K, Ali VM, Budgett RG, Ljungqvist A, Mountjoy M, Pitsiladis YP, Soligard T, Erdener U, Engebretsen L. IOC consensus statement: dietary supplements and the high-performance athlete. Br J Sports Med. 2018 Apr;52(7):439-455. doi: 10.1136/bjsports-2018-099027.

Stecker RA, Harty PS, Jagim AR, Candow DG, Kerksick CM. Timing of ergogenic aids and micronutrients on muscle and exercise performance. J Int Soc Sports Nutr. 2019 Sep 2;16(1):37. doi: 10.1186/s12970-019-0304-9.

Reviewer 2 Report

The work presented by the authors seems to me of great scientific interest. However, there are some questions that should be resolved in order to accept the manuscript.
At first glance, what first draws attention is the large number of authors in a study that, although there are 4 moments, there are only 8 athletes used. There are authors who indicate that any researcher who has not contributed at least 15% to the study should be grateful. I make this reflection for you because the inclusion of authors for convenience devalues ​​the work of those who have really worked.

Abstract

The abstract is very well oriented, however, the authors should include how long before the tests they ingested nitrates and in what form (capsules, juice, ...).
In what weight category did the athletes compete? Were they all the same?
If the objective of the study is to determine the effect of nitrates on performance, for what purpose are cognitive tests included? Wouldn't it be better to include these in the objective?
Introduction
Although the content of the introduction is adequate, I consider that the authors should summarize the first 2 paragraphs in 1 single paragraph. They should also justify better because they consider that NO3- could be suitable for athletes (perhaps using content from the 1st paragraph).
They should also justify why they use the lactate parameter as a performance test if nothing has been explained previously.
Justify its use as parameters such as lactate, RPE and cognitive tests. For what purpose do they use these parameters and why do you think they may be influenced by nitrate supplementation.
Material and methods
All athletes belonged to the same weight category?
In the inclusion criteria it is not clear what type of ergogenic aids they should not use during the previous 3 months. It seems like a long time for elite athletes. In the same way, how much training time per week did the athletes do? Did they all perform the same routines throughout the study?
It could be a problem of mine, but I don't understand figure 1 well. Likewise, it should be explained in the text how long before the physical tests they took the supplementation.
Nothing is said about the diet of athletes. They should take into account that the amount of carbohydrates taken pre-exercise or during exercise could modify the levels of lactate in the blood. In this sense, the authors should include the diet that the athletes followed during the experiment.
What software was used for the graphics? Was it the SPSS or another?
Discussion
Do the authors think that perhaps a chronic dose of nitrates (for example 10 weeks) could have modified the results? There are studies that show that a diet high in nitrates maintains the levels of NO for a longer time, which makes it easier to achieve all the functions of this. Discuss the possibility that the failure to obtain effects may be due to the short term used.
I miss a section that talks about the limitations of the study, its strengths and future lines of research.
Likewise, I miss a section of practical applications. How would they act in your athletes at least to improve RPE?
Conclusion
Improve this conclusion with more specific data?

Author Response

The work presented by the authors seems to me of great scientific interest. However, there are some questions that should be resolved in order to accept the manuscript.
At first glance, what first draws attention is the large number of authors in a study that, although there are 4 moments, there are only 8 athletes used. There are authors who indicate that any researcher who has not contributed at least 15% to the study should be grateful. I make this reflection for you because the inclusion of authors for convenience devalues ​​the work of those who have really worked.

Author Responds: Many thanks for your point. However, the study was not going to be performed with this amount of participants and we were unlucky to lose 4 participants, but this lost is usual in studies that include elite athletes. This study has been performed on a difficult context and without any funding. This only has been possible thanks to this team work which includes a list of authors whose implication and contributions have been essential (from the designing to the writing of the final draft).

Abstract

The abstract is very well oriented, however, the authors should include how long before the tests they ingested nitrates and in what form (capsules, juice, ...).

Author Responds: Thank you, we have added two cases that you have mentioned.

Lines 41-42: Participants consumed two doses of BJ-400 and BJ-800 or nitrate-depleted PL at 2.5 h prior to performing the Multiple Frequency Speed of Kick Test (FSKT).

In what weight category did the athletes compete? Were they all the same?

Author Responds: No, participants were in each category (<63 kg and <68 kg).

line 38-40: Eight trained male taekwondo athletes into two categories <63 kg (n=2) and <68 kg (n=6) (age: 20±4 years, height: 180±2 cm, body mass: 64.8±4.0 kg) completed four experimental trials using a randomized, double‐blind placebo‐controlled design: BJ-400, BJ-800, PL, and CON.

If the objective of the study is to determine the effect of nitrates on performance, for what purpose are cognitive tests included? Wouldn't it be better to include these in the objective?

Author Responds: we have revised it based on your suggestion

Lines 35-37: This study aimed to investigate two acute doses of 400 mg of NO3- (BJ-400) and 800 mg of NO3- (BJ-800) on taekwondo-specific performance tests and cognitive function compared with a placebo (PL) and control (CON) conditions.

Introduction
Although the content of the introduction is adequate, I consider that the authors should summarize the first 2 paragraphs in 1 single paragraph. They should also justify better because they consider that NO3- could be suitable for athletes (perhaps using content from the 1st paragraph).

Author Responds: We have summarised these two paragraphs. Currently, it exists only one paragraph.

They should also justify why they use the lactate parameter as a performance test if nothing has been explained previously.

Author Responds: Blood lactate concentration is an indicator of the contribution of glycolysis to the energy metabolism. Therefore, we don not consider blood lactate concentration as a performance variable. We only have assessed blood lactate concentrations as an indicator of the contribution of glycolysis to the energy metabolism. Based on this aim and the absence of any effect of BJ supplementation on lactate concentration on the previous studies, we have not added information in the introduction section. Nevertheless, if you consider, we could include information relative to this aspect.

Justify its use as parameters such as lactate, RPE and cognitive tests. For what purpose do they use these parameters and why do you think they may be influenced by nitrate supplementation.

Author Responds: Based on your suggestion, it has been extended the central effect of BJ.

Material and methods

All athletes belonged to the same weight category?

Author Responds: lines 139-141: we revised this

Therefore, eight athletes (age: 20 ± 4 years; height: 180 ± 2 cm; body mass: 64.8 ± 4.0 kg) two categories <63 kg (n=2) and <68 kg (n=6) completed the study.

In the inclusion criteria it is not clear what type of ergogenic aids they should not use during the previous 3 months. It seems like a long time for elite athletes. In the same way, how much training time per week did the athletes do? Did they all perform the same routines throughout the study?

Author Responds: Our participants were selected from Iran national league, so they were professional and had different levels from elite athletes. They were not members of the national team of Iran..

lines 141-143, 148-151: All athletes were from the same club to prevent potential interference/variation caused by different training programs implemented among taekwondo clubs. This study was conducted during the 4-week pre-competition preparation phase. During this time, participants trained six sessions per week, including three taekwondo-specific training and three condition sessions, including strength and taekwondo-specific training

Lines 143-147:  inclusion criteria included: 1) more than five years of experience in taekwondo; 2) not have consumed any ergogenic aids including; creatine, beta-alanine, caffeine, sodium bicarbonate, and  caffeine content supplements three months before the study.

It could be a problem of mine, but I don't understand figure 1 well. Likewise, it should be explained in the text how long before the physical tests they took the supplementation.

Author Responds: Based on this suggestion, we have added additional information and have enhanced the clarity of the Figure 1 as suggested by reviewer 1.

Nothing is said about the diet of athletes. They should take into account that the amount of carbohydrates taken pre-exercise or during exercise could modify the levels of lactate in the blood. In this sense, the authors should include the diet that the athletes followed during the experiment.

Author Responds: we added a sentence along with a reference.

Lines 219-225: Participants were also instructed to avoid drinks containing caffeine and nitrate-rich fruits and vegetables 72 h before the tests due to its potential ergogenic effect. Participants were asked to replicate their food and drink intake 24 h before each session. On the test day, water was provided ad libitum during the rest period between the tests and a standardized snack was ingested four hours before starting the test consisted of 1.5 g/kg carbohydrate and 20-gram protein

What software was used for the graphics? Was it the SPSS or another?
Author Responds: We added this information (lines 308-310)

Discussion
Do the authors think that perhaps a chronic dose of nitrates (for example 10 weeks) could have modified the results? There are studies that show that a diet high in nitrates maintains the levels of NO for a longer time, which makes it easier to achieve all the functions of this. Discuss the possibility that the failure to obtain effects may be due to the short term used.
I miss a section that talks about the limitations of the study, its strengths and future lines of research.

Author Responds: Thank you very much with your suggestions. We have included a paragraph that include the main limitations of this study and the future lines of research that must include a comparison between a chronic and an acute supplementation protocol.

Likewise, I miss a section of practical applications. How would they act in your athletes at least to improve RPE?

Author Responds: If it is possible that based on ES, our results imply practical applications. Therefore, we have revised it.

Conclusion
Improve this conclusion with more specific data?

Author Responds: We have improved conclusion with more specific data.

lines6 587-591: In summary, the results of this pilot study indicated that an acute BJ ingestion containing 400 mg of NO3- improved cognitive function. However, the ingestion of acute two doses of BJ supplementation containing 400 and 800 mg of NO3- in trained taekwondo athletes provided moderate and large effect sizes on anaerobic, but not statistical differences in taekwondo-specific performance.

Reviewer 3 Report

This study presents very interesting results, but the small sample size and large effect sizes for the aerobic and anaerobic variables of interest provide evidence that a significant difference could not have been captured with a sample size of 8 athletes. Therefore, the study should be reworked as a pilot study. Additionally, a discussion of the practical significance (if any) of the difference between variables that were not statistically significant but had large effect sizes is warranted.

Abstract

Lines 30-31:“The dose-response effect on taekwondo of BJ supplementation are yet to be determined.” Please either replace “of” with “following” and “are” with “is” or change “effect” to “effects”.

Lines 39-41:“Blood lactate was collected before the CMJ tests, immediately, and 3 minutes after the FSKT and PSST, with rating of perceived exertion (RPE) was recorded during and after specific taekwondo tests.” Please adjust the second half of this sentence for clarity.

Line 43 repeats results that are in the previous sentence. The previous sentence indicated that there were no differences between conditions for PSTT and FSKT performance which were earlier defined as taekwondo-specific tasks.

Introduction

Please remove “The” from Line 50.

Citation is required for lines 54-56, it’s not clear that this is also from the Campos et al. (2012) study.

Lines 56-58, Campos et al. (2012) indicated that these percentages are the estimated total energy costs of the exercise contributed to the different systems, not the total energy released by these systems.

It’s not clear what is meant by “in relevant contexts” on line 63, please clarify.

Lines 63-67 please reword for clarity.

Line 71, do you mean inconsistent evidence?

Line 79: I don’t think the thought is finished here regarding N03-, on line 79 “shown to modulate various aspects”, various aspects of what?

Lines 82-85: In reference to NO3- supplementation, Domínguez et al. 2018 wrote, “Thus, these effects, although they potentiate oxidative phosphorylation, have no repercussions on glycolytic energy metabolism. Hence, as beetroot juice has no alkalizing effect supplementation with this product is unable to reduce acidosis…”. The mention of acidic conditions within lines 82-85 may mislead the reader to think that NO3- supplementation aids with acidic conditions when in fact this is not true according to Dominquez et al., and the benefit of NO3- in this context appears to be with the resynthesis of phosphocreatine. Please adjust the second half of this sentence so the focus is on the depletion of PCr that occurs with high-intensity intermittent exercise.

Line 96-Delete the word “and”

Lines 109-110, “However, high and low dosage of BJ supplementation would be increased compared to placebo (PL) and control (CO) conditions.” Please indicate what would be increased.

Methods

Line 113: Please add that these are male athletes

Line 118: Delete “athletes need to have all of the following”.. and simply put “inclusion criteria included”

Line 124: Capitalize The Sport Science Research Institute of Iran, if appropriate.

Figure 1 shows that the Omni 1-10 scale was used but line 218 indicates that the Borg 6-20 scale was used please adjust the image for RPE in Figure 1. I also suggest adding the words “lactate”, “RPE”, and “Hooper test” to the figure. It is unclear what the picture is including the computer with the heart. Please clarify.

Lines 154-163: Stay consistent and either use minutes or hours, but not both.

Line 157: What do you mean by “in a dose-response manner”? Clarify.

Line 167: Add “the” before 24 hours

Line 173: Add “food and drink” before “intake”

Line193-195: Clarify what you mean by “athletes were pending the test” and add the word “set” before “the pace performance”. What is a “distance interval”? Do you mean time?

Line 214: How were jumps measured with this app? Describe the procedure with more specific information.

Line 234: change “complains” to “complaints”

Line 239, please indicate that the normality of the residuals was checked.

Line 240, please indicate that several one-way repeated measures ANOVAs were used as a single repeated measures ANOVA is unable to assess all your variables of interest across four treatment conditions.

Line 248 Either an alpha of .05 was used to determine the statistical significance or statistical significance was set as p ≤ .05. Please reword this sentence so either option is clear.  

Figure quality is low. Consider adding more numbered ticks on the y axis.

Results

Line 254 based on the definition of small, medium, and large effects in the statistical analysis section (consider a citation of the primary author for quantifying effect sizes), a partial eta squared of .285 is a large effect, not a medium effect. This large effect paired with and a p-value of .069 provides evidence that your sample size was not sufficient to find a significant difference. You should address this in the discussion.

Line 260, Similar to my previous comment a large effect size but no significant difference likely indicates that there is an effect, but the sample size was too small to reach statistical significance.

Table 1 should be presented in landscape view so that it can be more easily interpreted. Also, all abbreviations should be defined in the table so it may stand alone from the text.

Table 4: Change to “Rating” of perceived exertion. I do not see the % of participants who reached the level.

Line 334, The “s” in “statistical” needs to be capitalized here.

Based on Figures 3 and 4 it appears that there may have been influential outliers in your data set, please indicate how you decided if participants were or were not influential outliers.

Normality was mentioned in the statistical analyses section but whether the assumption of normality was met is missing from the results section. Please indicate whether the residuals were normally distributed and if they were not, how did that change your analyses?

Discussion

Line 360, The sample size was small for repeated measure ANOVA, and given that large effects were observed it’s difficult to conclude that your results failed to support the original hypothesis. By discussing the magnitude of the effect sizes, you may be able to indicate if your findings are of practical importance since the sample size was too small to identify statistical significance. If the differences are of practical importance, please use the following manuscript to help frame your discussion of this doi: 10.4300/JGME-D-12-00156.1. If the differences are not of practical significance please discuss why you think this is.

Line 369: change the spelling of “analyzed”

Line 373: What is meant by “these paralleled results”? Perhaps contradicting?

Line 379: Change wording for “have used chronic strategy”. Also, wording concerns on lines 392-393, lines 401-402, line 441, line 481, line 500

It is acknowledged on lines 401-403, that since nitrate levels were not assessed it’s difficult to know what the change from baseline was and if the increase in nitrate was sufficient to stimulate benefits associated with NO3- supplementation.

Line 424: What might explain the lack of ergogenic response? This needs clarity.

Author Response

This study presents very interesting results, but the small sample size and large effect sizes for the aerobic and anaerobic variables of interest provide evidence that a significant difference could not have been captured with a sample size of 8 athletes. Therefore, the study should be reworked as a pilot study. Additionally, a discussion of the practical significance (if any) of the difference between variables that were not statistically significant but had large effect sizes is warranted.

Author Responds: Many thanks for your attention to our study and your carefully revision. We agree with you. We have reworked this study as a pilot study (you can see it in the new title). In addition, we have discussed about the possible effect of the little sample size on results and we have added information about the possible implications of our results based on the analysis of ES, attending to your suggestions.

Abstract

Lines 30-31:“The dose-response effect on taekwondo of BJ supplementation are yet to be determined.” Please either replace “of” with “following” and “are” with “is” or change “effect” to “effects”.

 Author Responds: It has been done. Lines 31-32

Lines 39-41:“Blood lactate was collected before the CMJ tests, immediately, and 3 minutes after the FSKT and PSST, with rating of perceived exertion (RPE) was recorded during and after specific taekwondo tests.” Please adjust the second half of this sentence for clarity.

 Author Responds We have revised. Line 44.

Line 43 repeats results that are in the previous sentence. The previous sentence indicated that there were no differences between conditions for PSTT and FSKT performance which were earlier defined as taekwondo-specific tasks.

 Author Responds: We have removed it. Line 50

Introduction

Please remove “The” from Line 50.

 Author Responds: It has been done. Line 58

Citation is required for lines 54-56, it’s not clear that this is also from the Campos et al. (2012) study.

Lines 56-58, Campos et al. (2012) indicated that these percentages are the estimated total energy costs of the exercise contributed to the different systems, not the total energy released by these systems.

 Author Responds: we have revised it. please check the line of 57-61

It’s not clear what is meant by “in relevant contexts” on line 63, please clarify.

 Author Responds: Line 72- “in relevant contexts” changed to “relevant in combat sports”

Lines 63-67 please reword for clarity.

 Author Responds: Done please check lines 78-109

Line 71, do you mean inconsistent evidence?

Author Responds: Based on your commentary and suggestions of reviewer 2, it has been summarized and condensed the two first paragraphs in only one paragraph.

Line 79: I don’t think the thought is finished here regarding N03-, on line 79 “shown to modulate various aspects”, various aspects of what?

We have revised it

Author Responds: Lines 88-94 : NO3- is a nitric oxide (NO) precursor and part of the nitrateNO3-nitrite (NO2-) NO-nitric oxide pathway [12], shown to modulate various aspects of physiological mechanismes, including skeletal muscle function such as glucose and calcium hemostasis, muscle contraction and mechanical efficiency [13]. Improvements in exercise capacity are suggested to occur due to a lowering of the cost of ATP to exercising muscles [14, 15], increased vasodilation [16], and mitochondrial respiration, as well as intracellular calcium handling [17].

Lines 82-85: In reference to NO3- supplementation, Domínguez et al. 2018 wrote, “Thus, these effects, although they potentiate oxidative phosphorylation, have no repercussions on glycolytic energy metabolism. Hence, as beetroot juice has no alkalizing effect supplementation with this product is unable to reduce acidosis…”. The mention of acidic conditions within lines 82-85 may mislead the reader to think that NO3- supplementation aids with acidic conditions when in fact this is not true according to Dominquez et al., and the benefit of NO3- in this context appears to be with the resynthesis of phosphocreatine. Please adjust the second half of this sentence so the focus is on the depletion of PCr that occurs with high-intensity intermittent exercise.

Author Responds: Excuse us because the text was confused. The text has been clarified and modified. please check lines 94-98

Line 96-Delete the word “and”

Author Responds: this part changed Based on suggestions of reviewer 2.

Lines 109-110, “However, high and low dosage of BJ supplementation would be increased compared to placebo (PL) and control (CO) conditions.” Please indicate what would be increased.

 Author Responds: Excuse us. We have indicated it.

Lines 132-134: However, high and low dosage of BJ supplementation would increased both cognitive function and exercise performance compared to placebo (PL) and control (CO) conditions.

Methods

Line 113: Please add that these are male athletes

Author Responds: Line 138- Done.

Line 118: Delete “athletes need to have all of the following”.. and simply put “inclusion criteria included”

Author Responds: It has been done (Line 144).

Line 124: Capitalize The Sport Science Research Institute of Iran, if appropriate.

Author Responds: It has been revised (lines 151-152).

Figure 1 shows that the Omni 1-10 scale was used but line 218 indicates that the Borg 6-20 scale was used please adjust the image for RPE in Figure 1. I also suggest adding the words “lactate”, “RPE”, and “Hooper test” to the figure. It is unclear what the picture is including the computer with the heart. Please clarify.

Author Responds: Thank you very much. Based on your suggestion, we have improved the quality of the figure to be clearer.

Lines 154-163: Stay consistent and either use minutes or hours, but not both.

Author Responds: It has been revised (Line 198-225).

Line 157: What do you mean by “in a dose-response manner”? Clarify.

Author Responds: We have clarified (Line 200).

Line 167: Add “the” before 24 hours

Author Responds: It has been done (Line 213 ).

Line 173: Add “food and drink” before “intake”

Author Responds: It has been done (Line 222).

Line193-195: Clarify what you mean by “athletes were pending the test” and add the word “set” before “the pace performance”. What is a “distance interval”? Do you mean time?

Author Responds: It has been revised. Please check Line 243-247

Line 214: How were jumps measured with this app? Describe the procedure with more specific information.

Author Responds: We added in lines 263-267: Athletes CMJ were recorded by the cellphone’s camera and My jump 2 app measured and calculate the CMJ components.  This test was performed prior to the conduct of each specific taekwondo tests. Three CMJ were executed with a period of 30 seconds rest between jumps, and the highest jump was taken for analysis.

Line 234: change “complains” to “complaints”

 Author Responds: It has been done (Line 287).

Line 239, please indicate that the normality of the residuals was checked.

 Author Responds: It has been added (Lines 294-297).

Line 240, please indicate that several one-way repeated measures ANOVAs were used as a single repeated measures ANOVA is unable to assess all your variables of interest across four treatment conditions.

 Author Responds: It has been indicated (Line 297)

Line 248 Either an alpha of .05 was used to determine the statistical significance or statistical significance was set as p ≤ .05. Please reword this sentence so either option is clear.  Figure quality is low. Consider adding more numbered ticks on the y axis.

 Author Responds: It has been done (Line 306)

Results

Line 254 based on the definition of small, medium, and large effects in the statistical analysis section (consider a citation of the primary author for quantifying effect sizes), a partial eta squared of .285 is a large effect, not a medium effect. This large effect paired with and a p-value of .069 provides evidence that your sample size was not sufficient to find a significant difference. You should address this in the discussion.

 Author Responds: We appreciate your opinion and we have attended to this proposal during all the results section.

Line 260, Similar to my previous comment a large effect size but no significant difference likely indicates that there is an effect, but the sample size was too small to reach statistical significance.

 Author Responds: We appreciate your opinion and we have attended to this proposal during all the results section.

Table 1 should be presented in landscape view so that it can be more easily interpreted. Also, all abbreviations should be defined in the table so it may stand alone from the text.

Author Responds: Based on your suggestion, it has been edited and improved it

Table 4: Change to “Rating” of perceived exertion. I do not see the % of participants who reached the level.

Author Responds: It has been revised.

Line 334, The “s” in “statistical” needs to be capitalized here.

 Author Responds: Done (Line 407)

Based on Figures 3 and 4 it appears that there may have been influential outliers in your data set, please indicate how you decided if participants were or were not influential outliers.

Author Responds: Lines 294-297: All results are reported as mean and standard deviation, Normality was checked by using graphing (Boxplots, histograms, and scatterplots), interquartile range to create outlier fences, and hypothesis tests (Grubbs’ test) and all data were normally distributed (P > 0.05).

Normality was mentioned in the statistical analyses section but whether the assumption of normality was met is missing from the results section. Please indicate whether the residuals were normally distributed and if they were not, how did that change your analyses?

Author Responds: Lines 294-297: All results are reported as mean and standard deviation, Normality was checked by using graphing (Boxplots, histograms, and scatterplots), interquartile range to create outlier fences, and hypothesis tests (Grubbs’ test) and all data were normally distributed (P > 0.05).

Discussion

Line 360, The sample size was small for repeated measure ANOVA, and given that large effects were observed it’s difficult to conclude that your results failed to support the original hypothesis. By discussing the magnitude of the effect sizes, you may be able to indicate if your findings are of practical importance since the sample size was too small to identify statistical significance. If the differences are of practical importance, please use the following manuscript to help frame your discussion of this doi: 10.4300/JGME-D-12-00156.1. If the differences are not of practical significance please discuss why you think this is.

Author Responds: We have discussed it considering your suggestions.

Line 369: change the spelling of “analyzed”

Author Responds: Done (Line 444)

Line 373: What is meant by “these paralleled results”? Perhaps contradicting?

Author Responds: it has revised (449).

Line 379: Change wording for “have used chronic strategy”. Also, wording concerns on lines 392-393, lines 401-402, line 441, line 481, line 500

 Author Responds: We changed all of them.

It is acknowledged on lines 401-403, that since nitrate levels were not assessed it’s difficult to know what the change from baseline was and if the increase in nitrate was sufficient to stimulate benefits associated with NO3- supplementation.

Author Responds: Lines 478-483. In different studies has been reported a correlation between the increase in plasma NO2- following NO3- supplementation and improved performance [Wylie,2013 ]. Therefore, the absence of assessing NO2- or NO levels on this study could be converted as a limiting factor because it is not possible to check if the NO2- levels was increased sufficiently for obtaining ergogenic effects.

Line 424: What might explain the lack of ergogenic response? This needs clarity.

Author responds: This sentence has been moved to line of 510,511 to make clearer the lack of ergogenic effects of BJ.

Round 2

Reviewer 2 Report

The authors have improved the manuscript. It now could be accepted if the edirtor consider.

Author Response

Dear reviewer,

Thank you very much for you attention and carefully revision of this manuscript. The manuscript has improved significantly based on your suggestions and commentaries. 

Reviewer 3 Report

Thank you for your careful and thorough responses to my requests for revisions. The manuscript has greatly improved.

One further suggestion for clarity is that in the abstract and methods section the athletes are split into weights below 63kg and above 68kg. These groups are not compared so it’s not clear why they are reported as two distinct groups. I recommend removing the information on the groups in the abstract/methods sections and just report the average body mass of the athletes.

Lines-52-54: The conclusion in the abstract does not reflect that since the study has a lack of statistical difference, it should be interpreted with caution. The moderate and large effect sizes observed suggest that a statistical difference might have occurred if the sample size was increased. The conclusion section in the main text nicely summarizes the findings of this study by indicating the moderate-to-large effect of BJ supplementation but that this supplementation did not lead to statistical differences in taekwondo-specific performance in the current study. I recommend rewording the conclusion in the abstract so that it is similar to what is stated in the conclusion of the main text.

Lines 102-108: The sentence starting with “In addition,” and ending with “fatigue development” is quite long. Please break this sentence into multiple sentences for clarity.  

Line 133: Please change “increased” to “increase”.

Line 565 remove “be” and add an “s” so that the sentence reads “… could mask ergogenic effects of BJ.”

Lines 567-568 “a trend for statistical differences in aerobic performance” Add (TTE) after "performance" in this sentence as that refers back more directly to the results section because there are several markers of aerobic performance.

Author Response

Thank you for your careful and thorough responses to my requests for revisions. The manuscript has greatly improved.

One further suggestion for clarity is that in the abstract and methods section the athletes are split into weights below 63kg and above 68kg. These groups are not compared so it’s not clear why they are reported as two distinct groups. I recommend removing the information on the groups in the abstract/methods sections and just report the average body mass of the athletes.

Authors: Many thanks, we added these based on other reviewer’s suggestion. Now we have deleted it.

Please check line 38 and 141

Lines-52-54: The conclusion in the abstract does not reflect that since the study has a lack of statistical difference, it should be interpreted with caution. The moderate and large effect sizes observed suggest that a statistical difference might have occurred if the sample size was increased. The conclusion section in the main text nicely summarizes the findings of this study by indicating the moderate-to-large effect of BJ supplementation but that this supplementation did not lead to statistical differences in taekwondo-specific performance in the current study. I recommend rewording the conclusion in the abstract so that it is similar to what is stated in the conclusion of the main text.

Authors: Many thanks for your carefully review. Based on your certain appreciation, abstract has been revised:

Line 53: It was concluded that acute intake of 400 and 800 mg of NO3- rich BJ reported a moderate to large effect size in anaerobic and aerobic, however not statistical differences was found in taekwondo-specific performance.

Lines 102-108: The sentence starting with “In addition,” and ending with “fatigue development” is quite long. Please break this sentence into multiple sentences for clarity.  

Authors: We have converted it into two separate sentences:

In addition, BJ supplementation could exert a central effect based on an increased level of tension and subjective vitality at the start of exercise [19, 20]and improved physical performance with the same [21, 22] or a lower [20, 23] rating of perceived exertion (RPE). It is mediated by a possible increase in brain perfusion or a diminution of metabolite accumulation that attenuate activation of III/IV muscle afferent feedback to the central nervous system and the potential for central fatigue development[23]

Line 133: Please change “increased” to “increase”.

Authors: It has been revised

Line 565 remove “be” and add an “s” so that the sentence reads “… could mask ergogenic effects of BJ.”

Authors: It has been revised

Lines 567-568 “a trend for statistical differences in aerobic performance” Add (TTE) after "performance" in this sentence as that refers back more directly to the results section because there are several markers of aerobic performance.

Authors: It has been revised.